# 6-Axis Stress Tensor Sensor Using Multifaceted Silicon Piezoresistors

**DOI:** 10.3390/mi12030279

**Published:** 2021-03-08

**Authors:** Kentaro Noda, Jian Sun, Isao Shimoyama

**Affiliations:** 1Department of Intelligent Robotics, Toyama Prefectural University, 5180 Kurokawa, Imizu, Toyama 939-0398, Japan; k_noda@pu-toyama.ac.jp; 2Department of Mechano-informatics, Graduate School of Information Science and Technology, The University of Tokyo, 7-3-1 Hongo, Bunkyo-ku, Tokyo 113-8654, Japan; sun@leopard.t.u-tokyo.ac.jp

**Keywords:** MEMS, 6-axis tensor measure, Si-piezoresistor, multidimensional doping

## Abstract

A tensor sensor can be used to measure deformations in an object that are not visible to the naked eye by detecting the stress change inside the object. Such sensors have a wide range of application. For example, a tensor sensor can be used to predict fatigue in building materials by detecting the stress change inside the materials, thereby preventing accidents. In this case, a sensor of small size that can measure all nine components of the tensor is required. In this study, a tensor sensor consisting of highly sensitive piezoresistive beams and a cantilever to measure all of the tensor components was developed using MEMS processes. The designed sensor had dimensions of 2.0 mm by 2.0 mm by 0.3 mm (length by width by thickness). The sensor chip was embedded in a 15 mm^3^ cubic polydimethylsiloxane (PDMS) (polydimethylsiloxane) elastic body and then calibrated to verify the sensor response to the stress tensor. We demonstrated that 6-axis normal and shear Cauchy stresses with 5 kPa in magnitudes can be measured by using the fabricated sensor.

## 1. Introduction

Long life-time large structures, such as bridges and tunnels, accumulate deformation and fatigue inside of them because of the stresses caused by the small vibrations belonging to small earthquakes, the movement of cars and so on. Whenever these deformations and fatigue become larger than the rigidity of the building materials, the buildings will be damaged, which cause large accidents. For example, in 2012, a ceiling panel collapse accident occurred in Chuo Expressway Sasago Tunnel in Japan [1]. Additionally, in 2018, the Morandi bridge (Polcevera viaduct) in Italy collapsed because of aging deterioration [2]. These examples show that the fatigue of building materials is a critical factor that causes large accidents. To prevent such accidents, a method for predicting the occurrence of fatigue fracture is required.

To detect the early symptoms of fatigue fracture of building materials, several types of noninvasive detection methods such as ultrasound diagnosis [3,4] and acoustic emission measurement [5,6] have been proposed and realized. Since these methods can find a quite small crack before it becomes serious damage to buildings, they are quite effective methods for collapse prediction. However, although these measurement methods can detect minute cracks before they growth to a large crack which causes a serious accident, they cannot predict the occurrence of the first minute cracks before they occur. The crack will occur when the stress inside the material become larger than its’ strength. Therefore, a method that can measure the stress distribution inside the material more directly is required to predict the occurrence of the first and small crack. If the stress tensor information inside the building materials could be monitored to provide an early warning, then accidents may be prevented before any crack will occur.

The detection of stress information inside an object is very useful to us and very important in our daily life. However, if the stress tensor sensor is too large, then it may damage the structure of an object. A sensor chip must be sufficiently small to minimize the effect on the target structure. The sensitivity of the sensor is also a significant factor in detecting slight variations in the stress tensor. In Cauchy stress tensor theory, there are nine components of the stress tensor at one point. To obtain the most complete stress tensor information at a point, a sensor of small size with high sensitivity that can measure all nine components of the tensor is expected for stress tensor detection.

Several studies focus on stress tensor sensors. They made efforts to measure the stress tensor with a small size sensor and detect as many components as possible of the stress tensor.

For example, Shohei Kiyota et al. presented the concept of a sensor using the capacitance method [7,8]. They proposed a stress tensor sensing cube that had a rectangular rigid body with symmetrically-located capacitive stress sensors. The dimensions of the cross-sectional area of the cube was 8 mm by 9 mm. The cube was placed at the center of a silicone rubber and calibrated. The dimensions of the silicone rubber was 3 cm by 3 cm by 3 cm. In principle, a capacitive sensor can measure all of the tensor components; however, the smaller the capacitive sensor is, the larger the noise. Different types of sensor chips have different miniaturization requirements. The difficulty of miniaturization of such capacitive sensor cubes, to decrease the effect on the measured material, is a challenge for sensors using the capacitance method.

To reduce the size of the sensor chip, sensors which use piezoresistors [9,10] and complementary metal-oxide semiconductor(CMOS)-based structures [11,12] were reported. Wailed A Moussa et al., reported a multi-axes stress tensor sensor which can detect five-directional components of the stress tensor by placing piezoresistive elements in multiple positions [9]. They also succeed to prevent the effect of temperature change by comparing the responses of several piezoresistors [10]. Julian Bartholomeyczik et al. exploited a sensor that used an octagonal n-well in a p-substrate and eight peripheral contacts enabling the current to be switched in eight directions rotated by an angle of π/4 [11]. By taking advantage of the piezoresistive behavior of single-crystal silicon, the measurement of all in-plane stress tensor components was successful. The superficial area of the sensor chip could be miniaturized to several thousand μm^2^. Stress sensitivities for the sensor of 2 × 10^−7^ (kPa)^−1^ were demonstrated in the research. Additionally, Benjamin Lemke et al. reported a polyvalent CMOS stress sensor [12] that could be used to detect five independent components of the stress tensor with temperature compensation. The sensor had a footprint of only 29 by 29 μm^2^. It combined different piezoresistive sensors to detect the three shear stresses, the in-plane normal stress difference, and the sum of the three normal stresses. For normal stress sensing, the sensitivity was approximately 5 × 10^−7^ (kPa)^−1^. For shear stress sensing, the sensitivity was approximately 1 × 10^−6^ (kPa)^−1^. However, these sensors succeed to miniaturize the sensor size, the placement of the sensing elements were limited that they cannot detect all of the components of the stress tensor.

Broadway et al., reported on a stress tensor sensing method which measures the stress tensors inside the diamond [13]. The reported method measures the nitrogen-vacancy defect caused inside the diamond by the stress apply by using an optically detected magnetic resonance (ODMS) method. This method enabled the measurement of all directional components of the stresses inside the diamond which highly spatial resolution. However, the method can be applied only to limited materials thus it is difficult to apply to the multiple materials used for the buildings.

As the previous literature presented above indicates, although many studies have made efforts to fabricate stress tensor sensors, there are still some existing problems such as meeting the two requirements of realizing a stress tensor sensor of small size and detecting all the components of the tensor at the same time.

In this paper, the development of a sensor consisting of surface-doped and sidewall-doped piezo-resistive beams to detect six independent components of the stress tensor, which can be used to determine all of the tensor components, is reported. The piezo-resistive beams were formed using a MEMS process with a 20 μm-thick Si layer as part of a silicon on insulator (SOI) wafer. The dimensions of the designed sensor chip were 2.0 mm by 2.0 mm by 0.3 mm (length by width by height). All of the stresses were determined by measuring the resistance change with the extension and compression of the doped beams. The sensor chip was embedded in an elastic body to evaluate its response to the stress tensor. In theory, the response of each group of beams is proportional to the magnitude of the applied stresses and we performed a calibration to confirm the sensor responses for each component of the stress tensor.

## 2. Sensor Design and FEM Simulation

### 2.1. Sensor Design

The Cauchy stress tensor that we aim to measure with our proposed sensor is symmetric;
(1)$$\sigma \;=\left[\begin{array}{ccc} \sigma_{x} & \tau_{xy} & \tau_{xz}\\ \tau_{yx} & \sigma_{y} & \tau_{yz}\\ \tau_{zx} & \tau_{zy} & \sigma_{z} \end{array}\right]$$
therefore, although the stress tensor at a point consists of nine components, only six of these components are independent, which consist of three orthogonal normal stresses and three orthogonal shear stresses [14]. The symmetric components are six stress components, *τ_zx_*, *τ_xz_*, *τ_zy_*, *τ_yz_*, *τ_xy_*, and *τ_yx_*. The relationships among the six shear stress can be expressed as Equation (2).
|*τ_zx_* |=|*τ_xz_* |, |*τ_zy_* |=|*τ_yz_* |, |*τ_xy_* |=|*τ_yx_* |(2)

The values of symmetric components are equal. The directions of the symmetric components are orthogonal. Taking advantage of the symmetry of the Cauchy stress tensor, we only need to measure 6 independent components to determine the 9 components of the stress tensor at a point.

Figure 1a shows a schematic of the design of the proposed tensor sensor, which operates on the aforementioned Cauchy stress tensor principle and piezoresistive theory. The proposed sensor chip consists of the sensing elements which have a piezoresistor only on its surface and the ones which has piezoresistors on both surface and sidewall. Since our sidewall doping method has a limitation that it cannot form the piezoresistor only to the sidewall but the piezoresistor will be also formed to the surface of the structure simultaneously. Therefore, each sensor element was designed and arranged so as to be particularly sensitive to stress in a specific direction. Thus, the specific directional component of the stress can be derived by comparing the outputs of several elements.

This sensor chip will be embedded inside the center of a silicone rubber cube. When the sensor chip is embedded inside the silicone rubber, the microsized piezoresistive beams formed inside the chip will follow the deformation of the silicone rubber. Thus, the 6-axis tensor stresses applied to the surface of the silicone cube can be detected by measuring the deformation of the cube with the piezoresistive beams. Only by measuring the resistance changes.

The principles to detect stresses with each type sensing elements are shown in Figure 1b,c. Figure 1b shows the principle for the surface-doped beam. Although the entire upper surface of the beam is doped, by depositing Au onto the beam surface, the resistance change of the target part can be obtained. The parts with Au deposited on them can be used as electrode pads. For the surface-doped beams, the width is larger than the thickness, so the beams are easier to bend in the direction perpendicular to the beam surface. Figure 1c shows the principle of the beam which has piezoresistors on its surface and the side wall. The sum of the resistance changes of the piezoresistors formed on both surface and side wall can be measured by using the Au wiring partly formed on the surface of the beam. As same as the case of the surface doped beam, the resistance change of the aimed point of the piezoresistor can be detected by changing the design of the Au wiring.

There are six groups of sensing elements on this sensor chip. Among the six groups of sensing elements, Beam A and C consist of one beam. Beam B, D, and E consist of a pair of beams. Cantilever F is a U-shaped cantilever type structure. A schematic of each sensing structure is shown in Figure 2. The 3-axis normal stresses and the 3-axis symmetrical pairs of shear stresses can be detected by embedding this structure inside the cubic silicone rubber. When stress is applied to an elastic body such as silicone rubber, the elastic body usually deforms uniformly, so the microstructure placed inside moves along the deformation, and distortion which deforms the elements, as shown in Figure 1b,c, does not occur. However, when a structure with much higher rigidity than silicone rubber is arranged inside it, the difference between the deformation between the elastic body and the rigid structure will occur a distortion at the surrounding of the rigid structure. The stresses applied to silicone rubber will be able to measure by arranging the sensing elements (such as Figure 1b,c) around the edge of the sensor chip and by measuring this distortion. Additionally, we placed silicon walls around some of the sensing elements to prevent the propagation of some of the stresses and enabled to measure the target directional component of the stress with that element. The details of relationship between the sensing elements and the stress measurement are shown in the following.

Beams A, B, and C are designed to measure x, z, and y directional normal stresses mainly. Since Beam A is formed on the side edge of the sensor chip as shown in Figure 1, it follows the deformation of silicone rubber caused by x and z directional normal stresses and the xy, yx symmetrical directional shear stresses. Because piezoresistors are formed on the surface and the side wall of the Beam A, these stresses can be measured simultaneously by the Beam A. Similarly to Beam A, Beam C detects the normal stresses applied in y and z directions and the yz, zy symmetrical directional shear stresses simultaneously. On the other hand, Beam B only follows the z axis deformation caused by z axis normal stress, thus z axis normal stress can be measured according to the resistance change of Beam B. Since silicon walls are formed around the Beam B, it prevents the propagation of the deformation caused by the xy and yx directional shear stresses. Thus, this design enables Beam B to detect only z axis normal stress. Since the xy and yx directional shear stresses will deforms Beam B

On the other hand, Beams D, E and Cantilever F are designed to detect shear stresses applied to the surface of silicone rubber. As same as Beam A and C, these sensing elements have piezoresistors on both its surface and the side wall, however because of the wall around the sensing elements as shown in Figure 2, Beam D, E, and Cantilever F do not follows the deformation that caused by x and y directional normal stresses. Since all 6 elements detect different combinations of normal and shear stresses, it is possible to separate the stresses in each axial direction by combining the output results. However, each sensing element has a stress direction which they mainly measure, as shown in Table 1. The dimensions of piezoresistors suitable for to measure target stress is verified in the next section.

In this paper, our objective is to measure almost static change of the stress tensor occurred inside the building. Therefore, we designed our sensor to measure the static stresses. In our previous research [15], we found out that the sub-micro size silicon structures embedded inside the silicone rubber can follow the rubber’s deformation with the speed about 100 Hz order. According to this result, we consider that our sensor is effective to measure the static deformation without the influence of the time dilation toward the static deformation of the silicone rubber.

### 2.2. FEM Simulation to Design the Sensor Dimensions

In our study, we propose three types of sensing elements formed by using; beam types mainly used to measure normal stresses (Beams A, B, and C), a pair of beams, mainly used to measure shear stresses (Beam D and E), and a cantilever type mainly used to measure the shear stresses (Cantilever F). To determine the effective doped area of each type of element and the size of the surrounding silicone rubber for the stress measurement, we performed a structure analysis using finite element method (FEM) simulation. The commercially available software COMSOL Multiphysics (COMSOL, Inc., Burlington, MA, USA) was used for the simulation. Each dimension of the sensing elements was defined as in Table 2. In this simulation, we supposed that the elements are formed on a handle Si layer 300 μm thick. Additionally, we supposed that polydimethylsiloxane (PDMS) was used as a silicone rubber to cover the sensor chip. The PDMS was formed in a cubic style, and the sensor chip was placed at its center. The Young’s modulus of Si was set to 210 GPa, and that of PDMS was set to 750 kPa.

First, to determine the effective doped area of the beams to measure normal stresses, we set the surrounding PDMS cube size as 15 mm^3^, and determined the compressed/extended area by applying a normal stress to the surface related to the target beam. A schematic image of the calculation of the influence of x direction normal stress *σ*_x_ is shown in Figure 3. In the case of the x direction normal stress simulation, we fixed the bottom surface of the sensor and applied a 5 kPa x direction normal stress to the top surface of the PDMS cube. Figure 3b,c shows the compressed/extended region around Beam A. According to this result, whenever a normal stress was applied to the sensor, the center 90 μm region of the side of the beam opposite to the side where the normal stress was applied was extended, and the 45 μm beam end regions were compressed. According to this result, we confirmed that x and y direction normal stresses can be detected by forming a piezoresistive layer on a side wall of the beam. However, the resistivity of the piezoresistor depends on its strain; thus, the positive and negative values of the resistance change will reverse at the center and both ends of the beam. If we measure the total piezoresistance change of the whole piezoresistor formed on the sidewall of the beam, then it will become approximately zero through the cancellation at the center and both edges. Therefore, in our design we covered the 45 μm regions of the top surface at both ends of the beam with electrodes in the case of Beam A and C, which enabled us to measure only the resistance change of the piezoresistors formed at the center 90 μm area of the beam to detect mainly the normal stresses applied in the x and y directions (Figure 4a). In the case of Beam B, we used a pair of beams that were doped in their surfaces to detect the z axis normal stress. The proposed design of the beams is shown in Figure 4b. As shown in the figure, the 45 μm length regions of the surface at the ends of one beam is covered with metal wiring and the center 90 μm length region of the surface of the other beam is covered with a metal layer. As mentioned above, when a normal stress is applied to the sensor, the center of the beam will be extended and the ends of the beam will be compressed. Thus, the positive/negative resistance changes will be reversed at the center and the ends of the beam. Therefore, the resistance changes related to the *z*-axis normal stress can be doubled by calculating the differences in the resistance changes caused at the center and the ends of the beam by using the wiring design shown in Figure 4b. This will enable the measurement of a small *z*-axis normal stress with high sensitivity. In this work, we propose to use a differential Wheatstone bridge circuit to measure the difference in the resistance changes of two beams, such as in Beam B, to double the sensitivity.

Second, to determine the appropriate piezoresistive region to measure the 6-axis shear stresses with three elements, Beam D, E, and Cantilever F, we simulated the strain produced on each structure surface. In this simulation, the sensor chip was placed in the center of a cubic PDMS structure. The size of the PDMS cube was set to 3, 5, 15, and 100 mm^3^. One surface of the silicone rubber cube was fixed, and 5 kPa shear stresses were applied on the opposite side of the cube.

Since Beam D and E have symmetric structures, we only discuss the deformation of Beam E to decide the appropriate design of the piezoresistor for detecting mainly the shear stresses applied in zy and yz directions on the silicone rubber cube.

Figure 5a shows the strain distribution around Beam E when shear stress was applied in the zy and yz directions. The PDMS size in this simulation was 3 mm^3^. As shown in this result, whenever shear stresses were applied, the center 90 μm region of the y+ sidewall of the beams was extended, and the 45 μm regions at the ends of the y+ sidewall of the beams were compressed. On the other hand, when we focused on the y- sidewall, it was compressed in the center 90 μm region and extended in the 45 μm regions at the ends of the beams. According to these results, in our design, we formed piezoresistors on the inner sidewalls of the two beams, as shown in Figure 5b, and covered the surfaces at the 45 μm ends of each beam with metal wiring. Similar to the case of Beam B, because the positive/negative strain induced in the 90 μm length center regions of the inner sidewalls of the pair of beams, where we aim to form piezoresistors, are opposite, we can double the sensitivities of the beams to zy and yz direction shear stresses by measuring the difference in the resistance changes of the pair of beams. This design is also applicable to Beam D.

The strain distribution around the cantilever type structure F caused by shear stresses applied in the xy and yx directions is shown in Figure 6a. Similar to the simulation in Figure 3, the size of PDMS was set to 3 mm^3^ in this calculation. As shown in this result, the end of the inner sidewall surface of the y+ side beam of Cantilever F, as indicated by the red arrow, will be extended by shear stresses in both the xy and yx directions. To measure the strain around this region, in our design, we formed piezoresistors on this inner sidewall of the y+ beam and covered the surface of the cantilever except for the 135 μm length area from the end of the y+ beam. This sensor design enables the detection of the strain around the end of Cantilever F caused by xy and yx direction stresses. Thus, the xy and yx direction stresses can be determined with the Cantilever F structure.

As these results show, the 6-axis shear stress components can be measured by using the design of the three sensing elements, Beam D, E, and F.

According to Cauchy’s law, when the measurement point is small enough to consider it as a point, the magnitudes of the shear stresses applied in two symmetric directions become the same, and the strain induced in each shear stress sensing element, Beam D, E, and F, should show the same deformation under each symmetrical stress. This means that one shear stress sensing element can indirectly measure the change in the stress from the deformation of the elastic body. However, this method is inaccurate because the symmetric direction components of the shear stresses would actually be applied to different surfaces, and the sensor chip is not small like a theoretical point.

For example, in the case of Beam B, when a shear stress of 5 kPa was applied on the cube in the *τ_zy_*-direction, the average strain of the doped part of one of the beams was 5.46 × 10^−6^; however, when the shear stress was applied in the *τ_yz_*-direction, the average strain of the doped part was 7.10 × 10^−6^. According to these results, the strain induced in Beam B differed by 29% even though the shear stresses were applied in symmetric directions, when the size of the surrounding PDMS was 3 mm^3^. This error can be decreased by increasing the size of the PDMS. Increasing the PDMS size means that the relative size of the sensor chip becomes small compared to the whole sensor structure including PDMS. However, a best size for the measured object does not exist; there is only the better size for the measured object. The change of error with increasing size of the PDMS cube is shown in Table 3. The size of the PDMS cube is represented by *a* (mm). *ε*_1_ is the average strain of one of the beams of Beam E when the stress was applied in the *τ_zy_*-direction, and *ε*_2_ is the average strain of one of the beams of Beam E when the stress was applied in the *τ_yz_*-direction. The relationship between *ε*_1_, *ε*_2_, and *η* is shown in Equation (3).
(3)$$\eta =\frac{\left|\varepsilon_{2}-\varepsilon_{1}\right|}{\left|\varepsilon_{1}\right|}$$

According to Table 3, when the size of the surrounding PDMS cube (= 100 mm^3^) is 50 times the size of a side of the sensor chip (= 2 mm by 2 mm by 0.3 mm), the error caused by using one shear stress to detect two symmetric shear stress components will be reduced to 1.0%. If the measured object is sufficiently large, then the error can be ignored. However, the 100 mm^3^ size is too large for our purpose, to embed the sensor inside building materials and detect the inner stresses. If the sensor size is too large, then it would make the building materials fragile when embedding the sensor inside the materials. By considering the tradeoff between the cross talk of the symmetric shear stresses and the influence on the building materials, we designed the size of the surrounding PDMS cube to be 15 mm^3^ and evaluated the size influence on the sensor fabrication and experimental results.

## 3. Sensor Fabrication

To form piezoresistive layers in multiple directions, we used surface and sidewall doping methods similar to those in our past studies [16,17]. Figure 7 shows bird’s eye and cross-sectional views of the fabrication process of our sensor chip.

We used a p-type doped silicon on insulator wafer as the base material. The top device surface of this SOI wafer was p-type single crystal silicon. The SOI wafer consisted of 3 layers. The device Si layer at the top of the wafer was 20 μm thick, the SiO_2_ layer in the middle was 1 μm thick, and the handle Si layer at the bottom was 300 μm thick. The crystal direction of the top layer of the SOI wafer was (1, 0, 0). 

In the fabrication process, first, we formed five holes for to the device Si layer using the inductively coupled plasma-reactive ion etching (ICP-RIE) method as shown in Figure 7a. These holes were used to form the sidewall-doped surfaces of the sensing beams. Second, the wafer was spin-coated with n-type dopant (OCD P-59230, Tokyo Ohka Kogyo Co., Ltd., Kanagawa, Japan). P-59230 was composed of 3 wt % P_2_O_5_ solved in a liquid solvent. Because OCD is a liquid type material, it can coat both the wafer surface and the sidewalls around the etched holes simultaneously by the spin-coating method. After coating the wafer with the n-type dopant, we formed an n-type piezoresistor by heating the wafer momentarily. We call this method the rapid thermal diffusion method [18]. By using this method, thinner than 100 nm thick piezoresistors can be formed on both the surface and sidewalls of the wafer. Third, we formed electrodes on the sensor surface by depositing Cr/Au layers with vapor deposition method. These layers were patterned using the lift off method. In this step, the etching holes were completely filled by spin-coating photoresist several times to protect the piezoresistors formed on the sidewalls from metal contamination. The thicknesses of the patterned Cr/Au layers were 3 nm and 50 nm (Figure 7b). Fourth, the device Si layer was etched using ICP-RIE again to form the 6 groups of beam type structures as shown in Figure 7c. In this process, the etched holes were again completely filled with photoresist, similar to in the lift-off step, to protect the piezoresistors formed on the side walls. Finally, the handle Si layer and the SiO_2_ layer were etched from the back side of the wafer using ICP-RIE and hydrofluoric acid vapor to release the beams, as shown in Figure 7d.

After fabricating the sensor chip, we bonded the chip to flexible wiring and embedded them inside a silicone rubber cube. The flexible wiring was composed of a Au layer formed on 100 μm thick polyimide. Since the wiring was composed of flexible and thin materials, it did not disturb the deformation of the surrounding silicone rubber. To embed this sensor chip inside the silicone rubber, we prepared a 7.5 mm thick silicone rubber sheet and placed the chip on it. Then, we poured 7.5 mm thick liquid silicone rubber on the chip/sheet and embed the chip in the center of the whole silicone rubber structure. In this step, we used PDMS (KE-106, Shin-Etsu Chemical Co., Ltd., Kawasaki, Kanagawa, Japan) as the silicone rubber. The mixture ratio of this PDMS and its curing agent (CAT-RG, Shin-Etsu Chemical Co., Ltd., Japan) was 10:1.

Photographs of the fabricated sensor chip and the sensor embedded inside PDMS are shown in Figure 8 and Figure 9. Figure 8 shows the fabricated 6-axis tensor sensor. The sensor shape was cubic, and its size was 15 by 15 by 15 mm^3^. Figure 9 shows the scanning electron microscope (SEM) images of the surface and sidewall doped piezoresistive beams formed on the Si chip. The size of one sensor chip was 2 by 2 mm^2^. The parts in light color are the areas where Cr/Au wiring was formed.

## 4. Evaluation of Sensing Characteristics

To evaluate the sensing characteristics of the fabricated sensor, we conducted the following experiments. We first calibrated the sensor sensitivities to 6-axis tensor stresses and then measured multi-axis stresses applied at an arbitrary angle by using the calibrated result. Because we used our fabricated amplifier circuit for these measurements, we also evaluated the noise of our fabricated circuit to make the estimation more accurate.

### 4.1. Experimental Setups

To apply normal and shear stresses to the fabricated sensor, we used the following three types of experimental setups shown in Figure 10 and Figure 11.

First, the experimental setup used to measure the response of the sensor cube to the normal stress components of the stress tensor is shown in Figure 10a. In this setup, one arbitrary surface was fixed on the stage and the stress was applied on the opposite side surface. To apply a uniform stress on the surface, a hard, light acrylic board was placed on the surface of the sensor where the stress was applied to the cube. The normal stress would indirectly transmit to the surface of the sensor cube through the board. A force gauge was used to ensure that a stable force was applied on the acrylic board. Figure 11a shows an example of measuring a normal stress applied in the *σ*_z_-direction. The upper surface and bottom surface are the relative surfaces of Beam B. When the calibration was conducted at the time of applying a normal stress in the *σ*_z_-direction, the bottom surface was fixed on the stage by adhesive, and the upper surface was under a uniform normal stress applied by the force gauge. The same method was used for other normal stress measurements.

Second, the experimental setup used to measure the response of the sensor the shear stress components is shown in Figure 10b. The bottom surface of the sensor cube was fixed on the stage of the test stand, and shear stress was applied on the upper surface. To apply a uniform shear stress on the surface, an aluminum board was placed on the upper surface of the sensor cube. When the sensor cube was calibrated for the shear stress, a 300 g standard weight was placed on the aluminum board to prevent the board from slipping. The board was pulled by a force gauge on a linear guide through a wire. The shear stress would indirectly transmit to the upper surface of the sensor cube through the board. Figure 11b shows an example of measuring a normal stress applied in the *τ*_zx_-direction. The upper surface and bottom surface are the relative surfaces of Beam D. The bottom surface was fixed on the stage, and the upper surface was under a uniform shear stress to conduct the shear stress measurement. The same method was used for other measurements.

Finally, we used the experimental setup shown in Figure 10c to apply stresses to the sensor at an arbitrary angle. The dip angle is defined as *θ* (in degrees) as shown in Figure 11c. To apply a uniform stress on the surface, the acrylic board was placed on the upper surface of the sensor cube. The stress at an angle *ψ* would indirectly transmit to the upper surface of the sensor cube through the board. A force gauge was used to apply a stable stress on the acrylic board. The normal force was applied to the acrylic board which was attached to the sensor surface, and separated in *σ*_z_ and *τ*_zy_ directions according to the stage angle *θ*.

During these experiments, we used the Wheatstone bridge type circuits shown in Figure 12 to measure the resistance changes of the Si-piezoresistive beams formed in the sensor chip. In the experiments, a *V*_in_ = +1 V voltage was supplied to the bridge circuits and the resistance change of piezo-resistive beams were indirectly detected by the change in the output voltage. Because of the high sensitivity of the piezo-resistive beams, the output voltage will be very small before it is amplified 1000 times for observation by an oscilloscope. The non-inverting input of the amplifier + *V*_G_ was +5 V, and the inverting input of the amplifier −*V*_G_ was −5 V. A passive electronic low-pass filter was inserted at the output to eliminate unwanted high frequency noise. The selected resistor *R*_L_ for this low pass filter was 820 Ω and the selected capacitor *C*_L_ was 0.1 μF. Noise with a frequency larger than 1.94 kHz can be cut off by this filter. The bridge resistor *R*_0_ used in the circuit was 2 kΩ.

For the structures consisting of one piezoresistive beam, the circuit shown in Figure 12a was used to measure the resistance change of the beam. The differential voltage Δ*V*_out_ can be found as
(4)$$\frac{\Delta V_{\mathrm{out}}}{V_{\mathrm{in}}}=\frac{\Delta R}{4R}\cdot G$$

For the structures consisting of one pair of piezo-resistive beams, the circuit shown in Figure 12b was used to measure the resistance change of the beams. In this circuit, the difference in the resistance of the two beams was detected. Therefore, the resistance change and the voltage change can be expressed as,
(5)$$\frac{\Delta V_{\mathrm{out}}}{V_{in}}=-\frac{\Delta R}{2R}\cdot G$$
where *G* is the gain of the amplifier used to amplify the voltage outputs from the bridge circuit.

### 4.2. Sensor Calibrations

To calibrate the fabricated tensor sensor, we applied normal and shear stresses by using the setups shown in Figure 10a,b.

Figure 13 shows the relationships between normal stresses applied in the x, y, and z directions and the resistance change rates of the 6 beams. The sensitivity of the *x*-axis normal stress detector (Beam A) was Δ*R*_x_/*R*_x_ = 1.06 × 10^−3^
*σ*_x_ (Figure 13a). This relationship was obtained by using least-squares regression of the sensor outputs. The coefficient of determination R^2^ was 0.9995. This output could be compensated for by the values of the stress–strain matrix after the experiment. Similarly, when y and z axis normal stresses were applied to the sensor, the outputs of the sensor were as shown in Figure 13b,c. The sensitivities of the y axis detector (Beam C) and z axis detector (Beam B) were Δ*R*_y_/*R*_y_ = 1.08 × 10^−3^
*σ*_y_ and Δ*R*_z_/*R*_z_ = 1.27 × 10^−3^
*σ*_z_. The coefficients of determination were 0.9993 and 0.9995 in each experiment.

Similar to the normal stress cases, Figure 14 shows the relationships between shear stresses applied in the xy, xz, and yz directions and the resistance changes of the beams. When the shear stress was applied in the xy direction, the resistance change for the *τ*_xy_ detector (Cantilever F) was proportional to the amplitude of the stress (Figure 14a). The resistance change rate of Cantilever F was Δ*R*_xy_/*R*_xy_ = 1.49 × 10^−3^
*τ*_xy_. As in the case of Figure 13, the sensitivity of Cantilever F was calculated by using least-squares regression of the sensor outputs. The coefficient of determination R^2^ was 0.996. Similarly, the responses of the beams sensitive to zx direction shear stress (Beam D) and zy direction shear stress (Beam E) are shown in Figure 14b,c. The sensitivities of each sensitive beam were Δ*R*_zx_/*R*_zx_ = 2.34 × 10^−3^
*τ*_zx_ and Δ*R*_zy_/*R*_zy_ = 2.20 × 10^−3^
*τ*_zy_. The coefficients of determination R^2^ for both experiments were 0.995.

According to these calibration results, we determined the transformation matrix *M*^−1^ that can be used to calculate the 6-axis tensor stresses from the sensor outputs. Ideally, the output of the beams, except for the beam corresponding to the stress applied direction, should be zero; however, the other beam groups also bend with the deformation of the elastic cube body. Therefore, we had to consider the outputs of not only the target beam but also all the other beams inside the sensor chip to figure out the actual stress.

The relationships between the applied stresses and the sensitivities of all beams, which were measured in the calibration experiment, are shown as the following matrix:(6)$$\left(\begin{array}{c} \Delta R_{A}/R_{A}\\ \Delta R_{B}/R_{B}\\ \Delta R_{C}/R_{C}\\ \Delta R_{D}/R_{D}\\ \Delta R_{E}/R_{E}\\ \Delta R_{F}/R_{F} \end{array}\right)=10^{-7}\left(\begin{array}{cccccc} 10.4 & 17.2 & 3.36 & 70.4 & -4.98 & -3.36\\ -2.55 & -12.8 & -3.44 & -4.40 & 2.19 & -2.21\\ -1.60 & 3.96 & 10.3 & -7.53 & -28.8 & 6.37\\ 1.72 & 1.26 & -2.36 & -22.1 & -4.90 & 4.26\\ 2.03 & -1.92 & -6.84 & 2.26 & 20.2 & -5.04\\ -20.3 & -15.3 & 6.26 & 39.8 & -43.8 & -15.5 \end{array}\right)\left(\begin{array}{c} \sigma_{x}\\ \sigma_{z}\\ \sigma_{y}\\ \tau_{zx}\\ \tau_{zy}\\ \tau_{xy} \end{array}\right)$$

The transformation matrix *M*^−1^ is the inverse matrix of the matrix determined in Equation (6); thus, it becomes,
(7)$$\left(\begin{array}{c} \sigma_{x}\\ \sigma_{z}\\ \sigma_{y}\\ \tau_{zx}\\ \tau_{zy}\\ \tau_{xy} \end{array}\right)=10^{5}\left(\begin{array}{cccccc} 4.94 & 13.5 & 31.1 & 2.08 & 39.3 & -2.46\\ -0.926 & -11.8 & -5.37 & 3.67 & -2.49 & 1.50\\ -2.39 & 2.04 & 24.3 & -18.7 & 23.7 & -2.62\\ 0.931 & 0.737 & -6.21 & -0.682 & -9.08 & -0.0915\\ -1.48 & -1.57 & -2.99 & -5.87 & -3.47 & -1.17\\ 0.0662 & 1.15 & -33.1 & 0.946 & -53.1 & -2.72 \end{array}\right)\left(\begin{array}{c} \Delta R_{A}/R_{A}\\ \Delta R_{B}/R_{B}\\ \Delta R_{C}/R_{C}\\ \Delta R_{D}/R_{D}\\ \Delta R_{E}/R_{E}\\ \Delta R_{F}/R_{F} \end{array}\right)$$

From previous research [19], the evaluation of the sensor structure can be analyzed based on the singular value of the transformation matrix. The singular value of a matrix can be expressed as Equation (8).
(8)$${\sum^{\;}}_{a0}=\mathrm{diag}\left[\sigma_{a1},\sigma_{a2},\sigma_{a3},\sigma_{a4},\sigma_{a5},\sigma_{a6}\right]$$
where $\sigma_{a1}\geq \sigma_{a2}\geq \sigma_{a3}\geq \sigma_{a4}\geq \sigma_{a5}\geq \sigma_{a6}\geq 0$. For the transformation matrix in Equation (7), the singular value can be calculated as,
(9)$${\sum^{\;}}_{a0}=10^{-7}\mathrm{diag}[40.2,\;33.6,\;11.1,\;11.1,\;7.01,\;4.10].\;$$

The ratio of the largest singular value and the smallest singular value is approximately 9.80. For an ideal sensor, all the singular values are the same, and the ratio of the largest singular value and the smallest singular value is 1. With an increase in the ratio, the measurement error induced by the transformation will be larger.

We consider that there are two factors that reduce the sensing independence from the stress direction: Fabrication errors and the limitation of the size of the PDMS surrounding the sensor chip. According to the definition of Cauchy’s law, the stress tensor is defined as being caused at a point. However, because of the limitations of the application and fabrication, the size of the PDMS was limited to 15 mm^3^ when the sensor chip was 2.0 mm by 2.0 mm by 0.3 mm. This size limitation was the main reason for the error. We will evaluate the sensing characteristics by measuring the stress applied at an arbitrary angle according to this calibration result in the following experiment.

### 4.3. Circuit Noise Evaluation

To detect a small change in the resistance, the output was amplified 1000 times by an amplifier circuit. In this paper, we targeted to measure almost static change of the stress tensor for building fatigue evaluation. To measure this static stress tensor without high frequency noise, we used low pass filter for the output signal of each beam group. In the actual experiment, the high frequency electrical noise, which comes from the outer electronics and the power source, will be overlaid to the sensor output. Our low pass filter was designed to cutoff these noises and to increase S/N ratio of the sensor output. To evaluate the characteristics of this filter, we measured the noise level and compared it with the level of the static sensor output signal by the method introduced in previous research [20,21,22]. A sketch of the experimental setup is shown in Figure 15. In this experiment, +1 V DC power was supplied to the sensor cube as in the calibration experiment. However, there was no stress applied on the sensor chip at the time of detecting the noise level. The amplified output of the sensor cube was imported into the noise level analysis machine net analyzer to conduct the analysis.

Because the high frequency noise has been cut off by the low pass filter connected to the amplifier, the analysis will focus on the low frequency noise level. In this research, the noise level detected for each beam group from 1 Hz to 500 Hz is shown in Figure 16. The noise level analysis is based on a range of frequencies. For this experiment, we arbitrarily selected frequency ranges of 1–10 Hz and 1–100 Hz to analyze the noise level for each beam. The noise level for each beam in the two frequency ranges is shown in Table 4. From the experimental results, the noise levels in the frequency ranges of 1–10 Hz and 1–100 Hz are almost the same, of an order of 3 μV. If the output signal of the sensor chip is smaller than the noise level, then the effect of noise cannot be ignored, as shown in Figure 17.

The effect of the noise on the calibration experiment results was analyzed. We detected the minimum detectable Δ*R*/R from the noise ratio. Moreover, we calculated the minimum detectable stress in the relative direction from the detectable Δ*R*/*R* of the sensing elements. According to the sensitivity matrix calculated in the previous section, the minimum detectable stresses *σ*_0_ for each beam were calculated and are shown in Table 5.

### 4.4. Multiaxis Tensor Stress Measurement

In this experiment, we applied stresses at an angle, where the included angle between the stage and the sensor cube was 20, 30, and 40 degrees as shown in Figure 10c and Figure 11c. The stress *ψ* applied was from 0 to 10 kPa. The stress was applied on the top surface by pressing the acrylic board attached on the sensor surface with digital force gauge, and the bottom surface was fixed on the stage.

When the stress is applied at an angle, the stress can be considered as the synthesis of two stresses, one named *ψ*(*σ_z_*) that is perpendicular to the top surface and another named *ψ*(*τ_zy_*) that is along the top surface and parallel to the surface, shown in Figure 10c. The stress perpendicular to the top surface is in the *σ_z_*-direction, and the stress parallel to the surface is in the *τ_zy_*-direction.

Figure 18 shows the experimental results for the angle experiment. From the results of the angle experiment, we arbitrarily selected the data when *ψ* was 5 kPa, shown in Table 6. The transformation matrix was obtained in the previous section. Substituting the two stresses into the transformation matrix, the responses of each beam group can be calculated. The output data calculated with the obtained transformation matrix obtained are listed as Table 7. The responses obtained in the angle experiment and the responses calculated with the transformation matrix can be compared.

For the response of each beam, it is understood that although the orders of the experimental results and the results calculated with the transformation matrix are the same, their values differ and have large errors. In fact, when the stress was applied at an angle, the beams were bent due to the deformation of the PDMS elastic body. Additionally, because of the fabrication error, which means that the misalignment of the sensor chip from the center of the silicone rubber, the rotational deformation occurring inside the limited size silicone rubber cube affected the output of the sensor chip. For a small size elastic body, the effect of such stress and deformation at an angle is too large to ignore. However, when calculating the resistance change with the transformation matrix, the effect of the stress applied on the other beams was ignored, which led to the error in the responses of each beam.

## 5. Conclusions

A tensor sensor to measure six components of the stress tensor using six groups of piezo-resistive beams was developed. Each group of beams forms each detector for each stress tensor component. The *σ_x_*-detector and *σ_y_*-detector are both a one beam structure and sidewall doped. The *σ_z_*-detector, *τ*_zx_-detector, and *τ*_zy_-detector are all pairs of doubly supported beams. The *σ*_z_-detector is surface-doped. *τ*_zx_-detector and *τ*_zy_-detector are sidewall-doped. The *τ*_xy_-detector has U-shaped cantilever structure and is sidewall doped.

The actually fabricated sensor chip had dimensions of 2.0 mm by 2.0 mm by 0.3 mm (length by width by height). The sensor chip was embedded in a PDMS elastic body (15 mm by 15 mm by 15 mm), and the response to each component of the stress tensor was measured. The sensor response of each group of beams was proportional to the magnitude of the applied stress. The sensitivity of the *σ_x_*-detector is 1.06 × 10^−3^ (kPa)^−1^, the sensitivity of the *σ*_z_-detector is 1.27 × 10^−3^ (kPa)^−1^, the sensitivity of the *σ_y_*-detector is 1.08 × 10^−3^ (kPa)^−1^, the sensitivity of the *τ*_zx_-detector is 2.34 × 10^−3^ (kPa)^−1^, the sensitivity of the *τ*_zy_-detector is 2.20 × 10^−3^ (kPa)^−1^, and the sensitivity of the *τ*_xy_-detector is 1.49 × 10^−3^ (kPa)^−1^. This shows that the fabricated sensor chip has high sensitivity. With the transformation matrix obtained by calibration, six sensing structures could be used in the sensor to detect the six independent components of the stress tensor. Because there are only six independent tensor components at one point, the sensor can be used to determine all of the components of the stress tensor. The noise level of the sensing system is a slightly higher than 3 μV, at a small level. Comparing the noise level with the output signal of the sensor, for each beam group, if the applied stress is larger than 1.31~3.28 Pa (depending on the Beam), then the effect of noise in the ranges of 1–10 Hz and 1–100 Hz can be ignored.

However, in the case of the fabricated sensor, because of the limitation in the size of the surrounding silicone rubber, crosstalk existed between each beam. Therefore, we confirmed the measurement error when we measure the applied stresses and separate them into the directional components. According to Cauchy’s definition, for orthogonal stresses to be considered the same, the measurement points must be as small as possible. It was confirmed by FEM simulation that this also holds for the structure proposed in this study. We expect that the independence of the sensor can be improved when it is assembled inside a structure such as a bridge that is sufficient lager than the sensor chip size in practical use.

## Figures and Tables

**Figure 1 micromachines-12-00279-f001:**
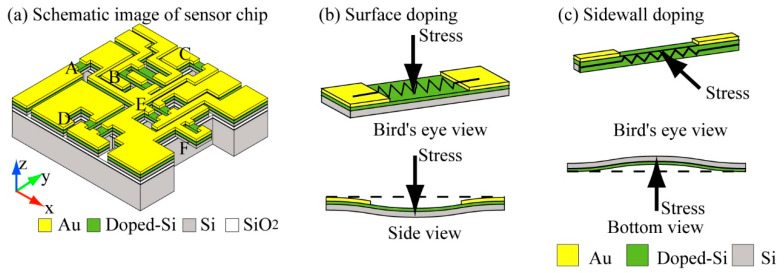
Schematic images of the proposed tensor sensor. (**a**) whole image of the sensor chip, (**b**) structure and deformation image of a surface doped silicon beam, and (**c**) structure and deformation image of the sidewall doped silicon beam.

**Figure 2 micromachines-12-00279-f002:**
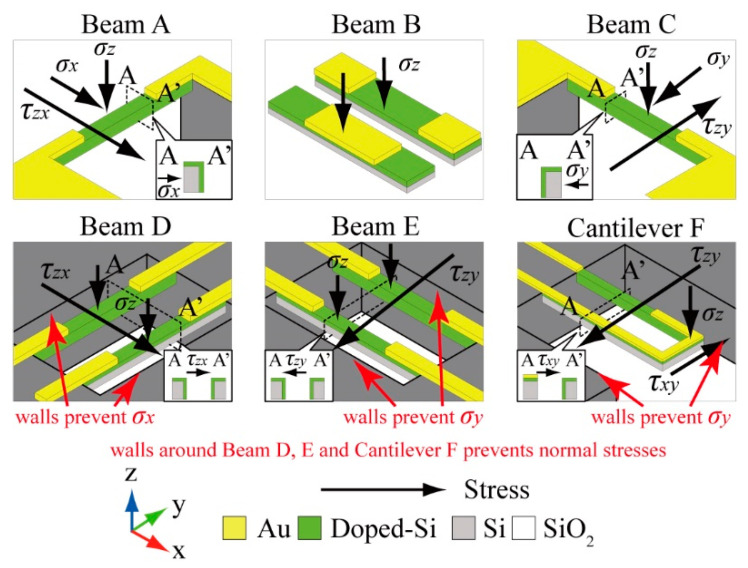
Schematic images of each sensing elements.

**Figure 3 micromachines-12-00279-f003:**
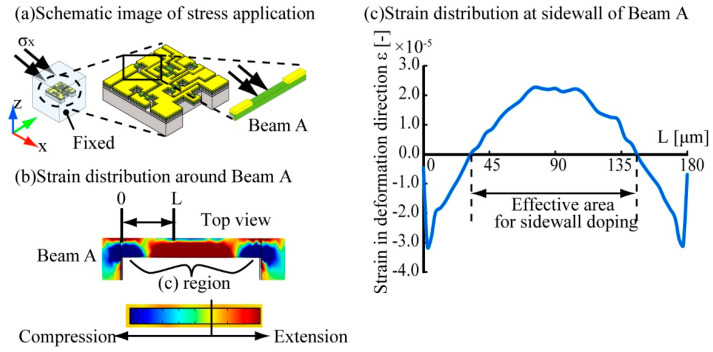
Finite element method (FEM) simulation of normal stresses. (**a**) schematic image of applying an x direction normal stress, (**b**) strain distribution around Beam A in response to *σ*_x_, and (**c**) strain distribution around the sidewall of Beam A.

**Figure 4 micromachines-12-00279-f004:**
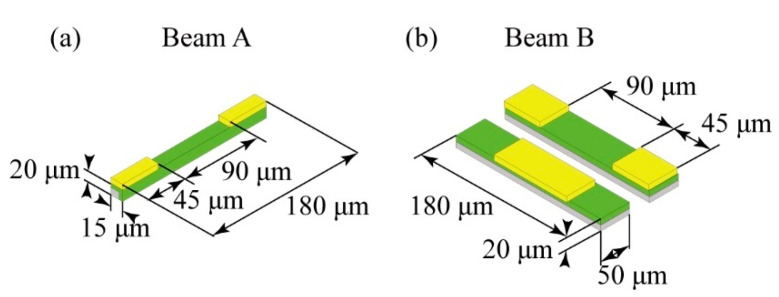
Design and dimensions of the normal stress sensing elements. (**a**) schematic image of the sensing element with piezoresistor on its side wall, (**b**) schematic image of the sensing element with piezoresistor on its surface.

**Figure 5 micromachines-12-00279-f005:**
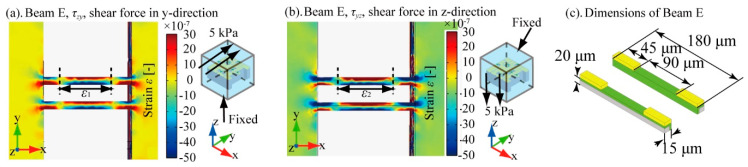
FEM simulation of shear stresses *τ*_zy_ and *τ*_yz_. (**a**) strain distribution around Beam E caused by *τ*_zy_, (**b**) strain distribution around Beam E caused by *τ*_yz_, and (**c**) sensor dimensions to detect shear stresses.

**Figure 6 micromachines-12-00279-f006:**
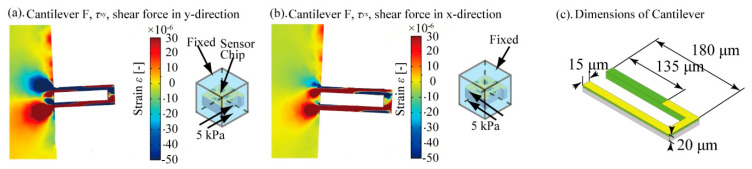
FEM simulation of shear stress *τ*_zy_ and *τ*_yz_. (**a**) strain distribution around Cantilever F caused by *τ*_xy_, (**b**) strain distribution around Cantilever F caused by *τ*_yx_, and (**c**) sensor dimensions to detect shear stresses.

**Figure 7 micromachines-12-00279-f007:**
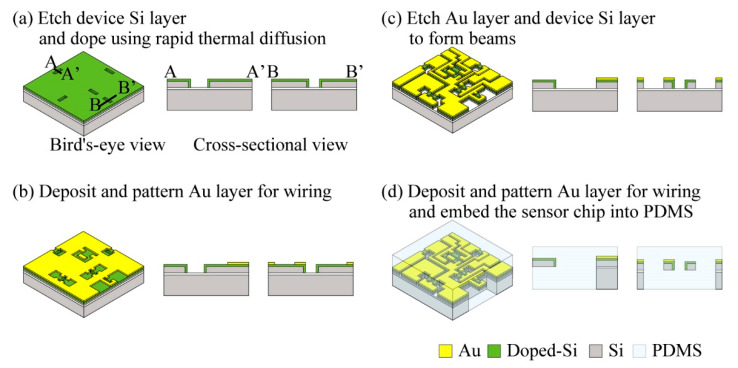
Overview of the fabrication process of the sensor. (**a**) step to fabricate piezoresistor on multiple walls, (**b**) step to form metal wiring, (**c**) Step to release sensing elements, (**d**) step to embed sensor chip inside silicone rubber (PDMS).

**Figure 8 micromachines-12-00279-f008:**
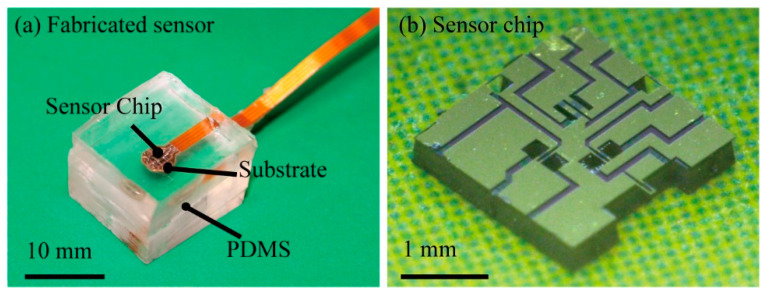
Photograph of the fabricated sensor. (**a**) whole image of the fabricated sensor and (**b**) sensor chip.

**Figure 9 micromachines-12-00279-f009:**
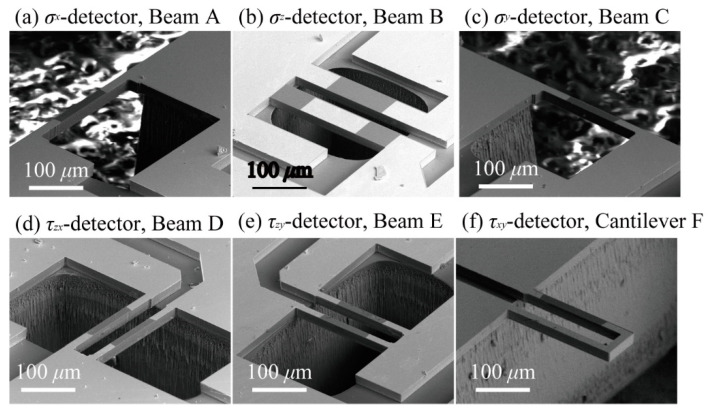
SEM images of the fabricated sensing elements inside the sensor chip. (**a**) SEM image of fabricated Beam A, (**b**) SEM image of fabricated Beam B, (**c**) SEM image of fabricated Beam C, (**d**) SEM image of fabricated Beam D, (**e**) SEM image of fabricated Beam E, (**f**) SEM image of fabricated Cantilever F.

**Figure 10 micromachines-12-00279-f010:**
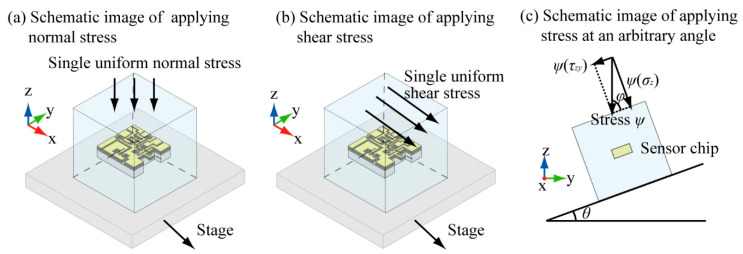
Schematic image of the experimental setups. (**a**) setup to calibrate normal stresses, (**b**) setup to calibrate shear stresses, and (**c**) setup to apply a shear stress at an arbitrary angle.

**Figure 11 micromachines-12-00279-f011:**
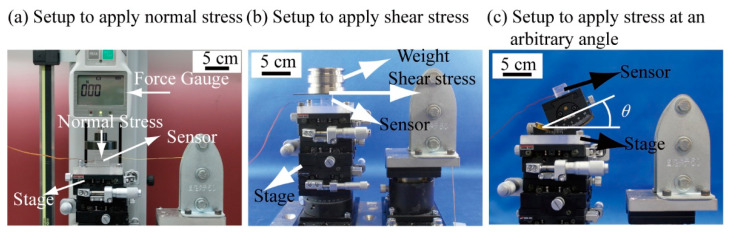
Photograph of the experimental setups. (**a**) setup to calibrate normal stresses, (**b**) setup to calibrate shear stresses, and (**c**) setup to apply a shear stress at an arbitrary angle.

**Figure 12 micromachines-12-00279-f012:**
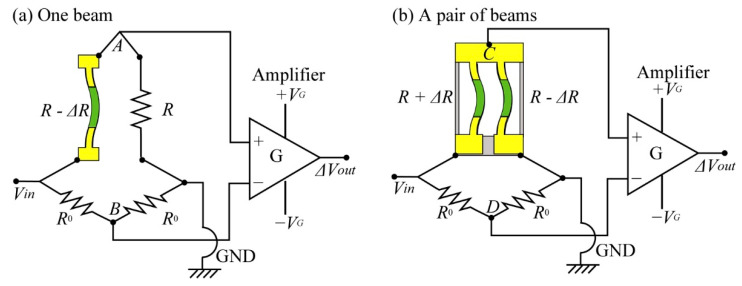
Schematic images of the Wheatstone bridge circuits used to measure the resistance changes of the sensor. (**a**) circuit to measure resistance change of a single element and (**b**) circuit for paired elements.

**Figure 13 micromachines-12-00279-f013:**
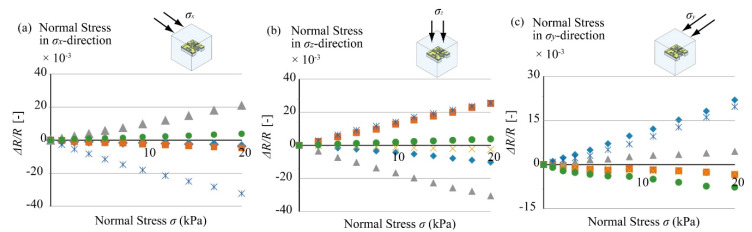
Sensor outputs under normal stresses; (**a**) responses to *τ*_zx_, (**b**) responses to *τ*_zy_, and (**c**) responses to *τ*_xy._

**Figure 14 micromachines-12-00279-f014:**
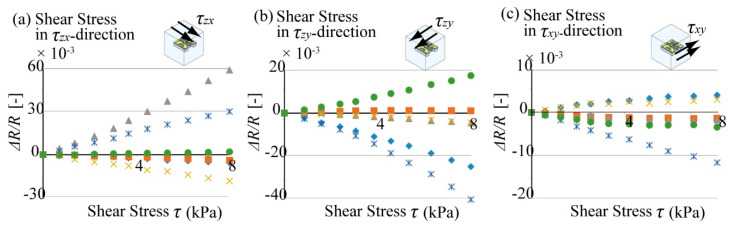
Sensor outputs under shear stresses; (**a**) responses to *τ*_zx_, (**b**) responses to *τ*_zy_, and (**c**) responses to *τ*_xy_.

**Figure 15 micromachines-12-00279-f015:**
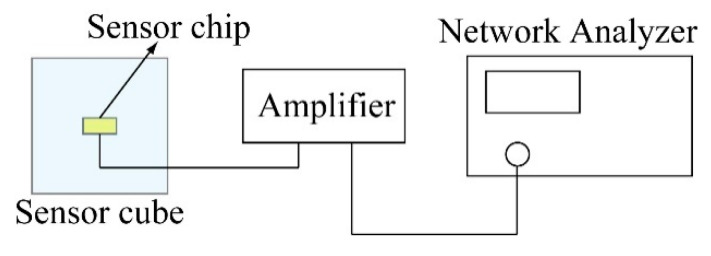
Schematic image of the setup to evaluate the noise level of the sensor.

**Figure 16 micromachines-12-00279-f016:**
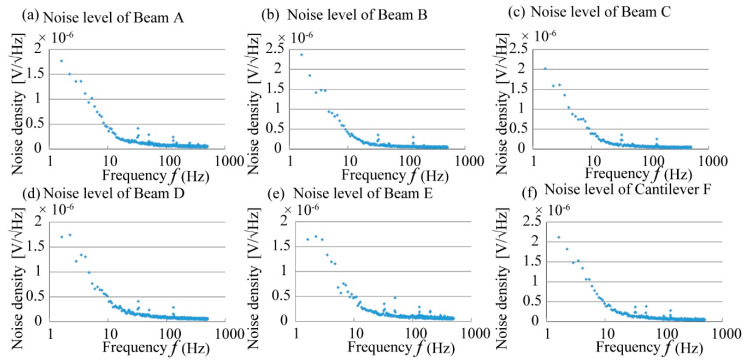
Noise levels of the sensing elements inside the fabricated sensor chip. (**a**–**f**) the measured results of the noise levels for each sensing element.

**Figure 17 micromachines-12-00279-f017:**
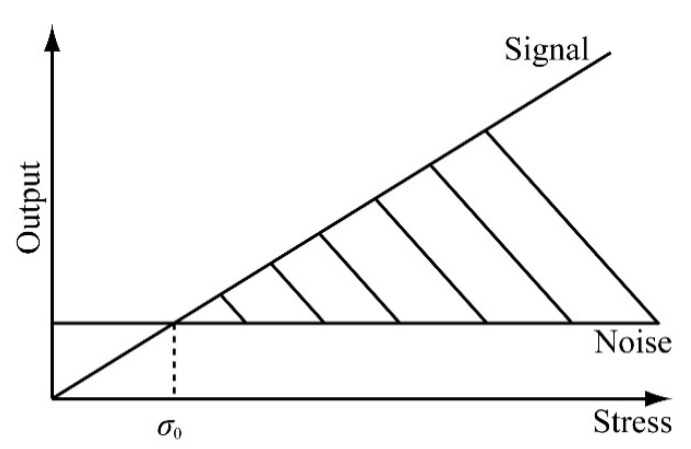
Relationship between the output signal and noise.

**Figure 18 micromachines-12-00279-f018:**
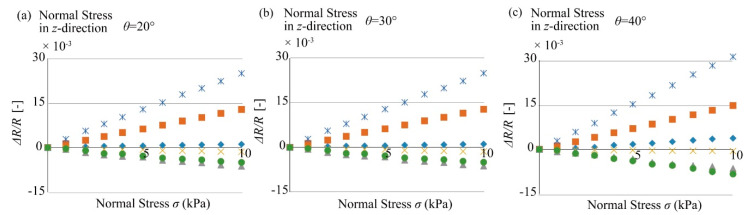
Sensor outputs toward shear stresses applied in 20, 30, and 40 degrees. (**a**) response of the sensor when it was tilted for 20 degree, (**b**) response of the sensor when it was tilted for 30 degree, (**c**) response of the sensor when it was tilted for 40 degree.

**Table 1 micromachines-12-00279-t001:** Relationships between doped surfaces and main target stresses.

Beam No.	Main Doped Surface	Main Target
A	Sidewall	*σ_x_*
B	Surface	*σ_z_*
C	Sidewall	*σ_y_*
D	Sidewall	*τ_zx_*
E	Sidewall	*τ_zy_*
F	Sidewall	*τ_xy_*

**Table 2 micromachines-12-00279-t002:** Dimensions of each sensing elements.

Beam No.	Length [μm]	Width [μm]	Thickness [μm]
A	180	15	20
B	180	50	20
C	180	15	20
D	180	15	20
E	180	15	20
F	180	15	20

**Table 3 micromachines-12-00279-t003:** Relationship between the size of the polydimethylsiloxane (PDMS) cube and the strains induced in the pair of beams for Beam E.

*a* [mm]	*ε* _1_	*ε* _2_	*η*
3	5.5 × 10^−6^	7.1 × 10^−6^	29%
5	4.9 × 10^−6^	6.1 × 10^−6^	24%
15	5.4 × 10^−6^	6.2 × 10^−6^	15%
100	7.7 × 10^−6^	7.6 × 10^−6^	1.0%

**Table 4 micromachines-12-00279-t004:** Noise levels of each element.

Beam No.	Noise Level [μV] 1–10 Hz	Noise Level [μV] 1–100 Hz
A	3.08	3.46
B	3.55	3.73
C	3.21	3.40
D	3.08	3.41
E	3.16	3.47
F	3.49	3.77

**Table 5 micromachines-12-00279-t005:** Minimum detectable stress.

Stress Direction	1–10 Hz		1–100 Hz	
	Δ*R/R* [×10^−6^]	*σ*_0_ [Pa]	Δ*R/R* [×10^−6^]	*σ*_0_ [Pa]
*σ* *_x_*	3.08	2.92	3.46	3.28
*σ* *_z_*	3.55	2.80	3.73	2.95
*σ* *_y_*	3.21	2.96	3.40	3.14
*τ_zx_*	3.08	1.31	3.41	1.45
*τ_zy_*	3.16	1.43	3.47	1.58
*τ_xx_*	3.49	2.34	3.77	2.52

**Table 6 micromachines-12-00279-t006:** Angle experiment results for ψ = 5 kPa.

Beam No.	Δ*R*/*R* (×10^−3^) (*θ* = 20°)	Δ*R*/*R* (×10^−3^) (*θ* = 30°)	Δ*R/R* (×10^−3^) (*θ* = 40°)
A	−3.41	−3.15	−3.39
B	6.43	6.59	7.30
C	0.617	0.881	1.81
D	−0.770	−0.453	−0.300
E	−2.70	−3.15	−4.03
F	12.9	13.8	15.7

**Table 7 micromachines-12-00279-t007:** Calculated results for ψ = 5 kPa.

Beam No.	Δ*R*/*R* (×10^−3^) (*θ* = 20°)	Δ*R/R* (×10^−3^) (*θ* = 30°)	Δ*R*/*R* (×10^−3^) (*θ* = 40°)
A	−7.23	−6.21	−4.99
B	5.64	5.00	4.20
C	3.07	5.49	7.75
D	0.244	0.677	1.09
E	−2.55	−4.21	−5.75
F	14.7	17.6	19.9

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
