# Peer review of "6-Axis Stress Tensor Sensor Using Multifaceted Silicon Piezoresistors"

_micromachines, 2021, doi:10.3390/mi12030279_

Round 1

Reviewer 1 Report

This paper presents the design, FEM simulations, fabrication and experimental results of a tensor sensor to measure six components of the stress tensor using six groups of piezo-resistive beams. The authors clearly explained the governing equations for the sensor design and for the experimental setup. The paper is well written and presented while its theoretical foundation has the required novelty. The present work’ connection with the open literature background is well and adequately justified. The paper presents the theoretical and simulations results and also an experimental validation. Further experiments can be also realized and improved. My specific comments are as follows: - The Introduction can be improved regarding the studies of fabricate stress tensor sensors. - Figure 1 a) please realize the corrections regarding the materials legend. - Page3, row 257, please correct the sentence mm3 in “…(=100 mm3)...”; the same rows 260 and 265, ...mm3... - Page 10, Figure 10, a and b captions...please correct the word Schematic - Page 11, rows 336-341, be consistent using Figure or Fig. along the paper text.

Author Response

Dear Reviewer, Thank you very much for your comments.

We are very glad you to review our thesis and giving us helpful comments to brush-up our paper. According to your comments, we revised our thesis as the followings.

(1) The Introduction can be improved regarding the studies of fabricate stress tensor sensors.  

Thank you for your comment. We added several but important references and revised our sentences in introduction to clarify the advantage of our work [9-10, 13]. The revised sentences were brush-upped in the introduction section.

Before: from line 60 to 61

To reduce the size of the sensor chip, CMOS based stress sensing principles were reported [9, 10]. 

After: from line 62 to 66

To reduce the size of the sensor chip, sensors which uses piezoresistors [9, 10] and CMOS based structures [11, 12] were reported. Wailed A Moussa et. al., reported a multi-axes stress tensor sensor which can detect 5-directional components of the stress tensor by placing piezoresistive elements in multiple positions [9]. They also succeed to prevent the effect of temperature change by comparing the responses of several piezoresistors [10].

Before: from line 71 to 73

However, these CMOS sensors succeed to miniaturize the sensor size, the sensing parts are concentrated in the upper surface of the sensor chip and cannot detect all the components of the stress tensor.

After: from line 77 to 85

However, these sensors succeed to miniaturize the sensor size, the placement of the sensing elements were limited that they cannot detect all of the components of the stress tensor.

Broadway et. al., reported on a stress tensor sensing method which measures the stress tensors inside the diamond [13]. The reported method measures the nitrogen-vacancy defect caused inside the diamond by the stress apply by using an ODMS (optically detected magnetic resonance) method. This method enabled to measure the all directional components of the stresses inside the diamond which highly spatial resolution. However, the method can be applied only to limited materials thus it is difficult to apply to the multiple materials used for the buildings.

(2) Figure 1 a) please realize the corrections regarding the materials legend.

Thank you for your comment, we fixed the Figure 1 as shown in our revised paper.

(3) Page3, row 257, please correct the sentence mm3 in “…(=100 mm3)...”;

the same rows 260 and 265, ...mm3...

Thank you for your comment, we changed the terms “mm3” to “mm3” in all sentences.

(4) Page 10, Figure 10, a and b captions...please correct the word Schematic

Thank you for your comment, we revised the term “Schematic” in Figure 10.

(5) Page 11, rows 336-341, be consistent using Figure or Fig. along the paper text.

Thank you for your comment, we revised the term “Fig.” in “Figure” to fit elsewhere in the paper.

Reviewer 2 Report

Dear authors, I have overall enjoyed the article. It embraces a full sensor construction from scratch, including simulation and experimental results. In general the article is OK, but some figures require some changes to make them better understandable. An extensive English review is also advisable. I next list some major changes that include presentation, English grammar and content corrections:

Line 13: The sentence “A sensor chip must be sufficiently small to minimize the effect on the building structure” has no relationship with the preceding sentence.

Line 38 : What do you mean in this sentence with “by using them”: “However, it is difficult to find the crack before it occurs by using them.” Previous sentence claimed that cracks could be detected, but in the following you say the opposite, please clarify.

As mentioned in the manuscript, the sensor was installed inside an encapsulating silicon rubber cube. Considering the dynamic behavior of the silicon rubber, how does it influences the accuracy of measurements for the resulting sensor? This is of special concern under conditions of dynamic loading.

Line 270: Could you provide more details of the Doped silicon used as the sensing mechanism of the piezoresistors, e.g. type and ratio of dopant, is silicon mono- or poly-cristaline?

Line 220: direction is misspelled.

Figure 10: The word stage is overlapped with an arrow, please correct it.

Figure 11: It is an advice to enlarge these photos; the experimental setup is an important part of the research.

Line 440. The sentence is too long, please consider splitting for a better understading: “An analysis of the minimum detectable ΔR/R and stress for each beam when the beam was under stress in the relative direction was conducted.”

Figure 18: please provide a legend for the markers on this figure.

In regard to Section 4.3; how does the filtering affect the dynamic response of force measurements? Later in Section 4.4, the static response of the sensor is assessed, but ultimately, this sensor is intended to operate under dynamic loading, or not? Please specify.

Line 445: In regard to the sentence “In this experiment, we applied stresses at an angle, where the included angle between the stage and the sensor cube was 20, 30 and 40 degrees.” How was this stress applied? Did you use the same setup previously described in Figure 11? Please specify.

In regard to Section 4.4; what about coupling between forces/torques applied in different directions? By coupling we understand the crosstalk occurring between a force applied in a given direction and the resulting measurement in a different direction. How can we assess coupling in the plots of Figure 18?

Author Response

Dear Reviewer, Thank you very much for your comments.

We are very glad you to review our thesis and giving us helpful comments to brush-up our paper. According to your comments, we revised our thesis as the followings.

(1) Line 13: The sentence “A sensor chip must be sufficiently small to minimize the effect on the building structure” has no relationship with the preceding sentence.

Thank you for your comment, we deleted the sentence to prevent confusion.

(2) Line 38 : What do you mean in this sentence with “by using them”: “However, it is difficult to find the crack before it occurs by using them.” Previous sentence claimed that cracks could be detected, but in the following you say the opposite, please clarify.

Thank you for your comments.

“them” inside the sentence indicates the previous measurement methods, such using AE and/or ultrasonic. These methods are effective to predict the occurrence of serious accidents because they can find the small early clacks. However, they cannot detect the crack before it occurs. On the other hand, if we can measure the stress distribution inside the material and can know the stress concentrating point, we may know where and when the crack will occur according to the magnitude of stress. This point is the advantage of our sensor to the previous methods. To clarify the point, we revised the introduction as the following.

Before: from line 34 to 41

To detect the early symptoms of fatigue fracture of building materials, several types of noninvasive detection methods such as ultrasound diagnosis [3, 4] and acoustic emission measurement [5, 6] have been proposed and realized. Since these methods can detect a quite small crack before it becomes serious damage to buildings, they are quite effective methods for collapse prediction. However, it is difficult to find the crack before it occurs by using them. To capture the progress of fatigue fracture, a method that can measure the stress distribution inside the material more directly is required. If the stress tensor information inside the building materials could be monitored to provide an early warning, then accidents may be prevented.

After: from line 33 to 43

To detect the early symptoms of fatigue fracture of building materials, several types of noninvasive detection methods such as ultrasound diagnosis [3, 4] and acoustic emission measurement [5, 6] have been proposed and realized. Since these methods can find a quite small crack before it becomes serious damage to buildings, they are quite effective methods for collapse prediction. However, although these measurement methods can detect minute cracks before they growth to a large crack which causes a serious accident, they cannot predict the occurrence of the first minute cracks before they occur. The crack will occur when the stress inside the material become larger than its’ strength. Therefore, a method that can measure the stress distribution inside the material more directly is required to predict the occurrence of the first and small crack. If the stress tensor information inside the building materials could be monitored to provide an early warning, then accidents may be prevented before any crack will occur.

(3) As mentioned in the manuscript, the sensor was installed inside an encapsulating silicon rubber cube. Considering the dynamic behavior of the silicon rubber, how does it influences the accuracy of measurements for the resulting sensor? This is of special concern under conditions of dynamic loading.

Thank you for your comment. In this paper, we targeted to measure almost static change of stresses which occur inside building material. Thus, we did not measure the time constant of our sensor in this paper. However, we agree to reviewer that the time constant of the sensor is one important point to evaluate the sensor.

In our past study, we found that the sub-um size silicon structure can follow the deformation of the deformation of the surrounding silicone rubber at 100 Hz order speed as shown in our new reference [15] added to the journal. From this result, we consider that our sensor has enough characteristics to measure the static deformation which is slower than 100 Hz. From this point, we added the next sentences to our paper to discuss about the time constant of our sensor.

Added sentence in line from 174 to 179

In this paper, our objective is to measure almost static change of the stress tensor occurred inside the building. Therefore, we designed our sensor to measure the static stresses. In our previous research [15] we found out that the sub-micro size silicon structures embedded inside the silicone rubber can follow the rubber’s deformation with the speed about 100 Hz order. According to this result, we consider that our sensor is effective to measure the static deformation without the influence of the time dilation toward the static deformation of the silicone rubber.

(4)Line 270: Could you provide more details of the Doped silicon used as the sensing mechanism of the piezoresistors, e.g. type and ratio of dopant, is silicon mono- or poly-cristaline?

Thank you for your comment and sorry to confusing you. In line 270, we are mentioning about the initial type of the SOI wafer, before forming the piezoresistor. The method/the material to fabricate piezoresistors on SOI surface is mentioned in line 276 in our first paper. In our experiment, we used p-type surface SOI wafer whose top surface was single crystal. Also the P-59230, the dopant we used to form piezoresistor on this SOI was composed of 3 wt% P2O5 according to the data sheet.

We revised these two sentences as following to prevent confusing and to add bit condition of the piezresistor as the following. Also we brushed-upped the revised sentences.

Before line 270

We used a p-type doped silicon on insulator (SOI) wafer as the base material.

After line from 288 to 289

We used a silicon on insulator (SOI) wafer as the base material. The top device surface of this SOI wafer was p-type single crystal silicon.

Before line 276

Second, the wafer was spin-coated with n-type dopant (OCD P-59230, Tokyo Ohka Kogyo Co., Ltd., Japan).

After line from 296 to 297

Second, the wafer was spin-coated with n-type dopant (OCD P-59230, Tokyo Ohka Kogyo Co., Ltd., Japan). P-59230 was composed of 3 wt% P2O5 solved in a liquid solvent.

(5)Line 220: direction is misspelled.

Thank you for your comment, we revised the typo.

(6)Figure 10: The word stage is overlapped with an arrow, please correct it.

Thank you for your comment, we revised the Figure.

(7)Figure 11: It is an advice to enlarge these photos; the experimental setup is an important part of the research.

Thank you for your comment, we doubled the sizes of the figures in Figure 11.

(8)Line 440. The sentence is too long, please consider splitting for a better understading: “An analysis of the minimum detectable ΔR/R and stress for each beam when the beam was under stress in the relative direction was conducted.”

Thank you for your comment, we separated the sentence as following. The revised sentence was brush-upped in the revised paper.

Before: from line 440 to 441

An analysis of the minimum detectable ΔR/R and stress for each beam when the beam was under stress in the relative direction was conducted.

After: from line 465 to 467

We detected the minimum detectable ΔR/R from the noise ratio. Also we calculated the minimum detectable stress in the relative direction from the detectable ΔR/R of the sensing elements.

(9)Figure 18: please provide a legend for the markers on this figure.

Thank you for your comment, we added the legends to the figure. Also we find the same trouble and revised Figure 13 and 14.

(10) In regard to Section 4.3; how does the filtering affect the dynamic response of force measurements? Later in Section 4.4, the static response of the sensor is assessed, but ultimately, this sensor is intended to operate under dynamic loading, or not? Please specify.

Thank you for your comment.

As mentioned in (3), we targeted to measure static stress in this paper. However, there were several high frequency noise appeared to the sensor output, which came from the outer environments such as electronics surrounding the experimental setup and the power source. Therefore, we had to take out these high frequency noises by using low pass filter. In Section 4.3., we evaluated the characteristics of this fabricate low pass filter and evaluated the S/N ratio of our sensor. To clarify this point, we revised the following sentences and brush-upped them.

Before: from line 423 to 427

To detect a small change in the resistance, the output was amplified 1000 times by an amplifier circuit. There was a low pass filter for the output signal of each beam group. In the actual experiment, the high frequency noise has been cut off but the low frequency noise will be amplified by the amplifier circuit. Therefore, we measured the noise level and compared it with the level of the sensor output signal by the method introduced in previous research [16-18].

After: from line 445 to 452

To detect a small change in the resistance, the output was amplified 1000 times by an amplifier circuit. In this paper, we targeted to measure almost static change of the stress tensor for building fatigue evaluation. To measure this static stress tensor without high frequency noise, we used low pass filter for the output signal of each beam group. In the actuatl experiment, the high frequency electrical noise, which comes from the outer electronics and the power source, will be overlaid to the sensor output. Our low pass filter was designed to cutoff these noises and to increase S/N ratio of the sensor output. To evaluate the characteristics of this filter, we we measured the noise level and compared it with the level of the static sensor output signal by the method introduced in previous research [16-18].

(11) Line 445: In regard to the sentence “In this experiment, we applied stresses at an angle, where the included angle between the stage and the sensor cube was 20, 30 and 40 degrees.” How was this stress applied? Did you use the same setup previously described in Figure 11? Please specify.

Thank you for your comment, in this experiment, we used the setup shown in Figure 10(c) and Figure 11(c). In these setups, we attached acrylic plate onto the sensor surface and applied normal force by using the digital strain gauge. Since the pitch angle of the stage was differed from 20 to 40 degrees, the applied normal force can be separated in σz and τzy directions according to the pitch angles. To clarify this point, we revised the sentence in line 445 as followings and brush-upped it.

Before: from line 338 to 339

The stress at an angle ψ would indirectly transmit to the upper surface of the sensor cube through the board. A force gauge was used to apply a stable stress on the board.

After: from line 358 to 362

The stress at an angle ψ would indirectly transmit to the upper surface of the sensor cube through the board. A force gauge was used to apply a stable stress on the acrylic board. The normal force was applied to the acrylic board which was attached to the sensor surface, and separated in σz and τzy directions according to the stage angle θ.

Before: from line 445 to 447

In this experiment, we applied stresses at an angle, where the included angle between the stage and the sensor cube was 20, 30 and 40 degrees. The stress ψ applied was from 0 to 10 kPa. The stress was applied on the top surface, and the bottom surface was fixed on the stage.

After: from line 471 to 474

In this experiment, we applied stresses at an angle, where the included angle between the stage and the sensor cube was 20, 30 and 40 degrees as shown in Figure 10(c) and 11(c). The stress ψ applied was from 0 to 10 kPa. The stress was applied on the top surface by pressing the acrylic board attached on the sensor surface with digital force gauge, and the bottom surface was fixed on the stage.

(12) In regard to Section 4.4; what about coupling between forces/torques applied in different directions? By coupling we understand the crosstalk occurring between a force applied in a given direction and the resulting measurement in a different direction. How can we assess coupling in the plots of Figure 18?

Thank you for your comment.

Because our sensor is composed of only 6 elements, it is difficult to divide moment/torque from the 6-axis stress tensor with current sensor chip. In the case of section 4.4, the sensor was attached to the stage and pressure σz and shear stress τzy were applied simultaneously and uniformly to the top surface of the sensor by using digital force gauge. We consider that, since the silicone rubber is an elastic material, the shear deformation caused by applied stress will not cause torque inside the rubber ideally. Also the sensor chip was designed to place at the center of the silicone rubber, thus the torque/moment will not effect the sensor output if the sensor was perfectly placed as it was designed. However, the fabricated sensor chip is too large compared to the surrounding silicone rubber to consider the chip as a point. Additionally, the Young’s modulus of silicone rubber is small, so that it will largely deform by the applied pressure/shear stress. Therefore, coupling of the shear stresses applied to the top surface of the sensor and its reaction stress occurred in the bottom surface will cause a torque inside the silicone rubber in the case of actual experiment. And this torque might affect the sensor chip inside the rubber. Thus, in the case of Figure 18 and Table 6 and 7, the measured result differed from the calculated ideal value. To clarify this point, we revised the following sentences.

Before: from line 460 to 462

In fact, when the stress was applied at an angle, the beams were bent due to the deformation of the PDMS elastic body. For a small size elastic body, the effect of such stress at an angle is too large to ignore.

After: from line 487 to 491

In fact, when the stress was applied at an angle, the beams were bent due to the deformation of the PDMS elastic body. Also because of the fabrication error, which means that the mis alignment of the sensor chip from the center of the silicone rubber, the rotational deformation occurred inside the limited size silicone rubber cube will affected the output of the sensor chip. For a small size elastic body, the effect of such stress and deformation at an angle is too large to ignore.

Round 2

Reviewer 2 Report

Mandatory changes have been done, the article can be published